# The Effects of Cannabidiol and Prognostic Role of TRPV2 in Human Endometrial Cancer

**DOI:** 10.3390/ijms21155409

**Published:** 2020-07-29

**Authors:** Oliviero Marinelli, Maria Beatrice Morelli, Daniela Annibali, Cristina Aguzzi, Laura Zeppa, Sandra Tuyaerts, Consuelo Amantini, Frédéric Amant, Benedetta Ferretti, Federica Maggi, Giorgio Santoni, Massimo Nabissi

**Affiliations:** 1School of Pharmacy, University of Camerino, 62032 Camerino (MC), Italy; oliviero.marinelli@unicam.it (O.M.); mariabeatrice.morelli@unicam.it (M.B.M.); aguzzicristina02@gmail.com (C.A.); laura.zeppa.95@gmail.com (L.Z.); giorgio.santoni@unicam.it (G.S.); 2Gynecological Oncology Department LKI, Leuven Cancer Institute KU, Leuven-University of Leuven, 3000 Leuven, Belgium; daniela.annibali@kuleuven.be (D.A.); sandra.tuyaerts@kuleuven.be (S.T.); frederic.amant@kuleuven.be (F.A.); 3School of Bioscience and Veterinary Medicine, University of Camerino, 62032 Camerino (MC), Italy; consuelo.amantini@unicam.it; 4Centre for Gynecologic Oncology Amsterdam (CGOA), Antoni Van Leeuwenhoek-Netherlands Cancer Institute (AvL-NKI), University Medical Centra (UMC), 1066 Amsterdam, The Netherlands; 5Oncologia Medica, Ospedale di San Severino, 62027 San Severino Marche (MC), Italy; benedetta.ferretti@sanita.marche.it; 6Department of Molecular Medicine, Sapienza University, 00155 Rome, Italy; federica.maggi@uniroma1.it; 7Integrative Therapy Discovery Lab, University of Camerino, 62032 Camerino (MC), Italy

**Keywords:** cannabidiol, TRPV2, endometrial cancer, progression-free survival, migration, chemo-resistance

## Abstract

Several studies support, both in vitro and in vivo, the anti-cancer effects of cannabidiol (CBD), a transient receptor potential vanilloid 2 (TRPV2) ligand. TRPV2, often dysregulated in tumors, is associated with altered cell proliferation and aggressiveness. Endometrial cancer (EC) is historically divided in type I endometrioid EC and type II non-endometrioid EC, associated with poor prognosis. Treatment options with chemotherapy and combinations with radiation showed only limited efficacy. Since no data are reported concerning TRPV2 expression as well as CBD potential effects in EC, the aim of this study was to evaluate the expression of TRPV2 in biopsies and cell lines as well as the effects of CBD in in vitro models. Overall survival (OS), progression-free survival (PFS), cell viability, migration, and chemo-resistance have been evaluated. Results show that TRPV2 expression increased with the malignancy of the cancer tissue and correlated with shorter PFS (*p* = 0.0224). Moreover, in vitro TRPV2 over-expression in Ishikawa cell line increased migratory ability and response to cisplatin. CBD reduced cell viability, activating predominantly apoptosis in type I cells and autophagy in mixed type EC cells. The CBD improved chemotherapeutic drugs cytotoxic effects, enhanced by TRPV2 over-expression. Hence, TRPV2 could be considered as a marker for optimizing the therapy and CBD might be a useful therapeutic option as adjuvant therapy.

## 1. Introduction

Medical Cannabis (MC) and pure phyto-cannabinoids (as tetrahydrocannabinol, THC and cannabidiol, CBD) have been largely studied for their ability to ameliorate some of the most common and debilitating symptoms in cancer patients, like nausea, vomiting and pain [1,2].

In addition to these effects, MC and THC/CBD have attracted attention as potential anti-tumoral drugs, mainly as adjuvant for current therapies in different type of cancers [3]. The main non-psychotropic cannabinoid compound, CBD, is present in the MC and is a potent inhibitor of different cancer properties, such as proliferation, migration and chemo-resistance in pre-clinical studies [3]. Anti-cancer effect of CBD is demonstrated in glioblastoma (GBM) [4,5,6], lung [7,8,9], gastric [10], breast [11,12,13], melanoma [14], prostate [15] and colon [16,17] cancers, neuroblastoma [18], multiple myeloma (MM) [19,20], leukaemia [21,22], pancreatic and endometrial cancers [23,24]. In addition, CBD has been shown to synergize with bortezomib and temozolomide in reducing cell growth and survival pathways in MM [19] and glioma cell lines [25], respectively.

It can exert its biological activities via dependent (D)- and independent (ID)-receptors mechanisms (RMs). D-RMs mainly include weak antagonistic effects on cannabinoid receptor type-1 and -2 (CB1, CB2) [26], agonist effects on some cation channels of the transient receptor potential (TRP) superfamily as vanilloid type 1–4 (TRPV1, TRPV2, TRPV3 and TRPV4), ankyrin 1 (TRPA1) and antagonist effects on TRP mammalian type 8 (TRPM8) [27,28]. Cannabidiol-related ID-RMs effects have also been described in cancer and are mainly based on evidences obtained in cell lines where the CBD receptors (CB1/CB2/TRPs) are not detected [20,29].

Regarding TRPV2, its expression and activation by CBD have been associated with deregulation of proliferation, cell differentiation and invasiveness in different cancer cell lines and animal models [30]. Briefly, TRPV2 over-expression was described in high-grade urothelial cancers’ biopsies and CBD-treatment induces apoptosis in bladder cancer cells [31,32]. In bladder and prostate cancers, TRPV2 stimulates migration and invasion [33,34,35]. Recently, it has been reported that TRPV2 should be considered a prognostic marker for triple negative and ERβ- breast cancer patients [13]. In addition, CBD induces a TRPV2-dependent differentiation and thus it sensitizes human glioma stem-like cells to chemotherapeutic drugs [4]. In MM, over-expressing TRPV2 cells are more susceptible to the effects of CBD [19].

Endometrial cancer (EC) is the most commonly diagnosed gynecological malignancy in developed countries. ECs are divided into two subtypes, type I endometrioid EC and type II non-endometrioid EC. Type I is well differentiated and is frequently associated with a hyper estrogenic environment. Type II carcinomas develop from atrophic endometrium and are classified in different histological subgroups (e.g., serous and clear cell adenocarcinomas). These subtypes are poorly differentiated and associated with dismal prognosis [36,37]. Treatment options are hysterectomy, chemotherapy, such as triplet therapy with paclitaxel (PAC), cisplatin (CIS), and doxorubicin (DOX), and combinations with radiation, but only limited options remain if the tumors relapse or metastasize [38]. Expression of CB1 and CB2 was evidenced to decrease in endometrial biopsies from women with EC compared to benign conditions, such as uterine prolapse, suggesting that CB receptors are potentially involved in progression of EC [39]. Thus, they should be useful as histological markers and therapeutic targets in the treatment or prevention of EC. In addition, expression of TRPV1 channels was demonstrated in EC cell lines and CBD was observed to induce cell death through TRPV1 only in a type I EC cell line [24]. Additional data on TRP channels have been reported in endometrial biopsies, where the levels of TRPV2, TRPV4, TRPM4, TRPM7, TRPC1, TRPC3, TRPC4, and TRPC6 channels were shown to fluctuate throughout the menstrual cycle in human endometrial stromal cells [40]. TRPV2, TRPV4, TRPC1/4, and TRPC6 were also present on biopsies derived from endometriosis patients; however, no correlation was found with endometriosis [41].

Since no data have been reported about TRPV2 expression in EC and there is a relative lack of robust diagnostic/prognostic biomarkers especially for EC type II, we firstly analyzed TRPV2 in human type II EC biopsies, and then, by in vitro investigations, we analyzed the TRPV2-dependent effects on cell migration and chemo-sensitivity. In addition, we evaluated the anticancer effects of CBD on EC cell lines.

## 2. Results

### 2.1. CB Receptors and TRPVs Gene Expression in EC Samples from TCGA

We assessed *CB* receptors and *TRPV1/2* gene expression in 506 EC data samples from TCGA, queried with cBioportal (TCGA, PanCancer Atlas). Samples were divided in type I endometrioid (397 samples) and type II serous type (109 samples).

In serous type samples, *CB1* receptor was highly expressed (*p* < 0.001), *CB2* was not expressed in both types. *TRPV1* and *TRPV2* were expressed in EC samples of both types. *TRPV1* was more expressed in serous subtype (*p* < 0.05) while *TRPV2* was more expressed in endometrioid subtype (*p* < 0.05) (Figure 1).

According to evidences in patients and since no data were available about TRPV2 and EC, we focused the attention on this channel.

### 2.2. TRPV2 Expression Increased with the Increasing of Non-Endometrioid Component

In order to evaluate the biological role of TRPV2 in EC, we measured the expression of TRPV2 in Ishikawa, MFE-280, HEC-1a and PCEM002 cell lines as type I EC models and PCEM004a and PCEM004b cell lines as mixed type I/II EC models, by RT-PCR and Western blot analysis. Results showed that all EC cell lines express low levels of *TRPV2* mRNA, although PCEM004a and b display a higher amount compared to the others (Figure 2A). We further analyzed if there was a difference between type I and mixed type cell lines by Western blot. Immunoblots demonstrated the TRPV2 protein expression only in mixed type I/II PCEM004 cells, and this expression increased with the increasing of non-endometrioid component (Figure 2B).

These results prompted us to investigate the correlation between TRPV2 expression levels and clinical parameters in a cohort of EC type II patients.

### 2.3. TRPV2 Expression Increased with the Malignancy of Type II EC and Correlated with a Shorter PFS

TRPV2 expression level was determined in a total of 68 cases, including serous, clear cell, mixed type, peritumoral tissues and normal endometrium. Expression data are summarized in Table 1 and Appendix A, divided for histological subgroups, International Federation of Gynecology and Obstetrics (FIGO) stage and age.

Representative images for staining and score are shown in Appendix A. Tissues were considered “high” at a score of 6 or higher, corresponding to weak staining in ≥67% of cells, moderate staining in ≥34% or strong staining in >10% of cells. “Moderate” staining was at a score of 4–5.

Highly expression of TRPV2 was found in 18.86% of tumor specimens, 30.19% were moderate and 50.94% stained low or negative. It was expressed in the epithelial component for 43.48% of specimens and in both stromal and epithelial components in 50% of samples. The highest score was most frequently detected in serous type (24.13%) while the lowest staining was found in clear cell type (71.43%). Regarding peritumoral and normal tissues, TRPV2 was predominantly moderate or low. Samples were classified according to FIGO staging system and we found that TRPV2 expression increases with increasing of tissue malignancy. No significant difference in TRPV2 expression was detected in patients ≤68 or >68 years, according to the median age. Furthermore, Kaplan–Meier analysis was performed, calculating overall survival (OS) and progression-free survival (PFS).

Kaplan–Meier analysis revealed that TRPV2 expression did not correlate with OS (TRPV2^high^ 37 months vs. TRPV2^moderate^ 53 months, *p* = 0.9346, HR = 1.039, 95% CI = 0.4131 to 2.615, TRPV2^high^ 37 months vs. TRPV2^low^ 43 months, *p* = 1.326, HR = 1.039, 95% CI = 0.5579 to 3.149, TRPV2^moderate^ 53 months vs. TRPV2^low^ 43 months, *p* = 1.326, HR = 1.199, 95% CI = 0.5665 to 2.537). High TRPV2 expression correlated with a shorter PFS (TRPV2^high^ vs. TRPV2^low^
*p* = 0.0224, HR = 4.675, 95% CI = 1.244 to 17.57, TRPV2^high^ vs. TRPV2^moderate^, *p* = 0.1172, HR = 2.755, 95% CI = 0.7754 to 9.790, TRPV2^moderate^ vs. TRPV2^low^, *p* = 0.6896, HR = 1.232, 95% CI = 0.4433 to 3.422) (Figure 3).

Additionally, OS and PFS were calculated according to TRPV2 distribution, dividing patients according to TRPV2 expression in tumor, stroma, or both (tumor + stroma). Stroma subgroup was excluded because there were too few patients for statistical analysis. Intra-tumoral TRPV2 distribution did not influence OS outcome (*p* > 0.05, HR = 0.6764, 95% CI = 0.3391 to 1.349) or PFS (*p* > 0.05, HR = 0.8102, 95% CI = 0.3201 to 2.051) (Figure 4).

Taken together, our data showed that TRPV2 expression correlates with malignancy in non-endometrioid EC and supported the investigation of CBD biological effects in EC.

### 2.4. TRPV2 Expression Stimulated Migration and Survival of EC Cells

According to data obtained in patients, we decided to investigate the role of TRPV2 on proliferation and migration of Ishikawa cells, which show low TRPV2 expression. Cells were firstly transfected with TRPV2 expressing vector (TRPV2^+^ cells) and TRPV2 expression was subsequently confirmed by Western blot analysis (Figure 5A). The result showed that TRPV2^+^ cells exhibited higher migratory capacities compared with untransfected cells, after 48 h, as determined by the wound-healing assay (*p* < 0.05) (Figure 5B).

Additionally, since Protein kinase B (PKB), also known as Akt, pathway is involved in supporting several pro-tumoral processes, to assess the potential role of TRPV2 in regulating cancer cell migration, modulation of AKT/PKB pathway was evaluated through Western blot analysis.

Indeed, AKT phosphorylated form was significantly increased in TRPV2^+^ cells (*p* < 0.001) supporting the hypothesis that TRPV2 expression increases cancer aggressiveness (Figure 6).

### 2.5. TRPV2 Expression Influenced the Effect of Chemotherapeutic Drugs

Untransfected and TRPV2^+^ Ishikawa cells were treated with CIS and DOX (from 0.0064 µg/mL to 100 µg/mL), and PAC (from 0.00025 µg/mL to 4 µg/mL) for 72 h. Results showed that TRPV2 expression increased chemo-sensitivity in Ishikawa cells for CIS (*p* < 0.05), while it did not particularly modulate the effect of DOX and PAC (Figure 7).

### 2.6. CBD Induced Cytotoxicity in EC Cell Lines in Single and Daily Administration

The effect of CBD, a TRPV ligand, in reducing cell viability was evaluated in EC cell lines in single and daily administration, washing cells with fresh medium in daily treatment. Cells were treated for 72 h with CBD (up to 15.72 µg/mL) and percentage of cell viability was evaluated by a cell viability MTT assay (Figure 8). The results showed a dose-dependent CBD effect in all EC cell lines, with IC_50_ values reported in Table 2. Additionally, daily administration induces higher cytotoxicity when compared with CBD single administration (Figure 8, Table 2). So, we decided to work with the doses of 3.92, 7.85 and 15.72 µg/mL, according with the IC_50_ of each EC cell line, in daily administration.

### 2.7. CBD Induced Cell Death in Type I EC Cell Lines

We investigated the cytotoxic effect of CBD in EC cells by Annexin V-fluorescein isothiocyanate (FITC) and Propidium Iodide (PI) staining. Each cell line was treated with an appropriate dose of CBD according to the IC_50_ in daily administration, up to 72 h. Then, fluorescence was analyzed by flow cytometry. The results evidenced that CBD treatment increased the percentage of Annexin V^+^ or Annexin V^+^/PI^+^ cells in MFE-280, HEC-1a and in primary PCEM002 cell lines, indicating apoptotic cell death, but only PI^+^ cells were detected in Ishikawa cell line, indicating a necrotic cell death (Figure 9). In PCEM004a and PCEM004b, CBD treatment resulted in a non-significant increase of PI and/or Annexin V^*^ cells. This result was not comparable with cell viability reduction demonstrated by MTT. For this reason, we decided to deepen investigate CBD effects in PCEM004a and PCEM004b mixed type EC cells.

### 2.8. CBD Induced Cell Cycle Arrest and Autophagy in Mixed Type I/II EC Cell Lines

In order to investigate the reduction in cell viability in mixed type I/II PCEM004a and PCEM004b, the role of CBD treatment in regulating cell cycle was evaluated. The results showed that CBD was able to induce cell accumulation in the G1 phase (Figure 10). These data evidenced that CBD-reduced cell viability in PCEM004 cells, was partially due to cell cycle inhibition.

Since it has been demonstrated that CBD is able to induce autophagy in cancer [4], we investigated whether the reduction of cell viability and the G1 cell accumulation by CBD treatment was due to an autophagic process in mixed type I/II primary cell lines. We analyzed the conversion of the soluble form of LC3 (LC3-I) to the lipidated and autophagosome-associated form (LC3-II), marker of autophagy activation after 48 h and 72 h of daily treatment, using Western blot analysis. We found that CBD induced a strong increase of the cleaved LC3-II form, especially in PCEM004a after 72 h (Figure 11A).

To confirm autophagy activation, we used acridine orange dye, which accumulates in acidic vesicular organelles, considered as an indicative marker of autophagy. These organelles emit bright red fluorescence, the intensity of which is proportional to the degree of the acidity and the volume of these structures, thus we assessed this fluorescence at the microscope. As depicted in Figure 11B, CBD treatment promoted acidic vesicular organelles formation after 48 h, especially at 7.85 µg/mL, in PCEM004 cell lines.

### 2.9. CBD Inhibited Migratory Ability of EC Cells

To further investigate anti-tumor effects of CBD, we treated Ishikawa, PCEM004a and PCEM004b cells with CBD for 24 h and then measured cell migration. The results showed that CBD was able to reduce migratory capacity of Ishikawa cells compared with vehicle-treated cells in a dose-dependent manner, with a significant effect at 7.85 µg/mL (*p* < 0.05) (Figure 12).

In both PCEM004 cells, CBD strongly reduced migratory ability of EC cells, already from the lowest doses of 3.92 µg/mL, up to inhibiting completely the migration at 7.85 µg/mL (*p* < 0.001) (Figure 13, Appendix A).

### 2.10. CBD Increased the Effect of Chemotherapeutic Drugs Used for EC Treatment

The effect of CBD combined with chemotherapeutic drugs was evaluated in TRPV2-transfected Ishikawa cells, compared with untransfected cells. Both models were exposed to CBD at 3.92 µg/mL, in combination with two non-cytotoxic doses of each chemotherapeutic drug: CIS 0.25 and 0.5 µg/mL, DOX 0.015 and 0.03 µg/mL, PAC 0.0015 and 0.003 µg/mL for 72 h. Chemotherapeutic drugs were administered once, while CBD was administered daily. The combined treatment induced an increased level of cytotoxicity, as compared with either CBD or all chemotherapeutic drugs alone, in both models (Figure 14). Additionally, the combination of chemotherapeutic drug and CBD, except for PAC, was significantly more cytotoxic in TRPV2-transfected cells (Figure 14). These results suggest that CBD enhanced the effect of chemotherapeutic drugs, used at lowest doses, in a TRPV2-dependent manner, thus supporting the feasible use of CBD as an adjuvant in the treatment of human EC.

## 3. Discussion

The non-endometrioid type II EC is responsible for most EC-related deaths because it is characterized by an aggressive behavior and high resistance to the common therapies. Furthermore, there are no targeted therapies approved so far for this subtype, and it is still treated in the same way as endometrioid type I EC, which is generally characterized by good prognosis and good response to therapy [42,43]. The relative lack of robust diagnostic/prognostic biomarkers for EC and its often-late presentation impedes the improvement of its morbidity and mortality rates. According to Taylor et al., the prognosis of EC is based exclusively on hormone receptor status that is considered not fully efficient at prognostic value [44]. This suggests that novel markers should be considered for clinical value [39]. Recently, Ayakannu et al. have shown that CB1 and CB2 receptors expression, at both transcriptional and protein levels, was reduced in human samples of both endometrioid and non-endometrioid EC [39]. These results are in contrast with previous studies of Guida et al. [45] and Fonseca et al. [24], in which was reported, respectively, that CB2 and CB1 are highly expressed in human samples and in two EC cell lines, type I Ishikawa and Hec50co, which expressed some characteristics of type II EC. Our data obtained from TCGA are in accordance with Ayakannu et al.’s findings. The reason for this could be related to methodological differences, as suggested by the authors [39]. Regarding TRPVs, Fonseca et al. demonstrated that TRPV1 was expressed in Ishikawa and Hec50co cell lines, but no data were provided for another CBD target, such as TRPV2 [24].

Herein we show that TRPV2 expression augmented with the increasing of a non-endometrioid component and its expression in human EC samples of type II and mixed type was associated with a reduced PFS. Additionally, its expression increases with malignancy, according to FIGO stage. Previously, it has been demonstrated that TRPV2 expression correlates with worse OS and PFS for triple negative breast cancer [13], esophageal squamous cell carcinoma [46], MM [47] and gastric cancer [48]. Furthermore, its expression increases with increasing of tumor stage in urothelial cancer [34], gastric cancer [48] and esophageal squamous cell carcinoma [46]. TRPV2-mediated signaling pathways modulate some pathological processes in cancer [30]. Indeed, previous evidences show that inactivation of TRPV2-mediated signals in cancer leads to an uncontrolled proliferation and cell death-resistance, and that TRPV2 activation increases migratory capability and invasiveness of cancer cells [49]. TRPV2 enhances migration process of prostate cancer cells through a TRPV2-mediated calcium influx, via Gq/Go, phosphatidylinositol-3,4 kinase (PI3,4K) signaling and translocation of TRPV2 from endosome to the plasma membrane [34]. Here, we show that TRPV2^+^ EC cells have a high migratory ability, and its expression induces an increase of pAKT/AKT ratio, supporting the activation of PI3K pathway in TRPV2^+^ cells that leads to high migratory ability and survival. These data could support the evidence that high expression of TRPV2 in patients correlates with shorter PFS.

High TRPV2 expression or its activation by CBD was associated to enhance drug-uptake with a resulting increase of chemosensitivity and cytotoxic effects on cancer cells in GBM and triple negative breast cancer [13,25,50]. Indeed, it has been demonstrated that DOX permeates directly through TRPV2 pore channel, in TRPV2-transfected GBM cells [25]. Herein, we show that, treating TRPV2^+^ EC cells with CIS, DOX or PAC, the presence of TRPV2 expression increased CIS cytotoxic effect, suggesting that TRPV2 can influence chemo-sensitivity to this drug. Compared with GBM cells, Ishikawa cells are more sensitive to DOX (~50-fold), and its increase in uptake and cytotoxicity TRPV2-dependent, as previously described in GBM [25], are not appreciable in our model. Moreover, as in MM and breast cancer cells [19,25,51], combination of CBD and chemotherapeutic drug, dispays an increased cytotoxic effect compared with chemotherapeutic drug or CBD alone, especially in TRPV2-transfected cells, suggesting that this effect could be partially due to TRPV2 activation by CBD. Overall, TRPV2 expression in EC could be potentially considered as a marker to optimize the therapy.

Up to now, the antitumoral effect of cannabinoids in different human cancer cell lines and in vivo preclinical models has been demonstrated [3,4,5,6,7,8,9,10,11,12,13,14,15,16,17,18,19,20,21,22,23,24]. In GBM, CBD alone and in combination with THC, induced a cell viability reduction and apoptosis in vitro and in vivo [25,52,53]. Regarding lung cancer, CBD induced apoptotic cell death in in vitro model and in primary cells derived from patients; moreover, it induces tumor regression in in vivo model [7]. In breast cancer, CBD inhibited AKT and mTOR signaling inducing autophagic cell death [12]. In MM, CBD was able to reduce cell proliferation and induce a necrotic cell death [19]. Our results evidence as CBD was able to reduce cell viability in EC cell lines, inducing predominantly apoptotic cell death. These data are in agreement with previous findings related to the efficacy of CBD as anticancer molecule.

Recently, Fonseca et al. published preliminary data regarding anticancer effect of eCBs, THC and CBD in Ishikawa and Hec50co cell lines. They found that a significant reduction in cell viability was observed with eCBs and CBD in both EC cell types, while, with THC treatment (0.01–25 μM), no changes were observed. In Ishikawa cells, CBD at 5 µM (1.57 µg/mL) induced chromatin condensation, increased levels of ROS/RNS, decreasing mitochondrial membrane potential, which results in an apoptotic cell death [24]. These data are quite in contrast with our findings. Indeed, CBD induces apoptotic cell death in MFE-280, HEC-1a, and PCEM002 but in Ishikawa cell line, CBD treatment results in increased PI^+^/Annexin^−^ cell population. The main difference is probably related with CBD concentration used and treatment protocol. Indeed, in our study, cells were treated daily and a necrotic cell death in Ishikawa cell line was detected with a higher concentration of CBD compared with Fonseca et al.’s experiments. Additionally, our data demonstrate that CBD treatments leads to cell cycle arrest and autophagic process in the mixed type I/II cells. Autophagy is an intracellular process by which cellular material is delivered to lysosomes for degradation allowing self-renewal and survival but, in cancer, autophagy has opposing context-dependent roles and then, autophagy stimulation or inhibition has been proposed as cancer therapy [54]. It has been found that CBD is able to induce autophagy, as demonstrated in breast cancer, colon rectal cancer and glioma stem-like cells (GSC), leading to cell death, increased chemosensitivity and inhibition of GSCs’ proliferation and clonogenic capability [4,12,55]. These data support our results in which CBD induced an autophagic process in the most aggressive and resistant mixed type cells, which express TRPV2 channel, thus, arrested cell proliferation and migration. However, since it is known that CBD acts both in a dependent (D)- and independent (ID)-receptors mechanisms (RMs) [26,27,28], our results support that CBD effects can be obtained also in TRPV2-low expressing EC cells.

## 4. Materials and Methods

### 4.1. The Cancer Genome Atlas (TCGA) and cBioportal Database Analysis

The cBioPortal for Cancer Genomics is an open-access downloaded bio-database, providing visualization and analyzing tools for large-scale cancer genomics data sets (http://cbioportal.org). Analysis of 506 sequenced EC samples from this database (PanCancer Atlas) was performed in order to evaluate PD-Ls expression, following the online instructions of cBioPortal database for Genetic Alteration, Mutation level, Clinical Attribute, and mRNA expression. Briefly, the cancer-specific TCGA datasets were selected followed by selection of mRNA expression z-scores relative to all samples (log RNA Seq V2 RSEM) and, *CB1*, *CB2*, *TRPV1* and *TRPV2* gene symbols, in the specified columns. On submitting the query, the software shows all types of genomic alterations including somatic mutations, copy number change, and mRNA expression, in a concise graphical summary called oncoprint. Then, data were downloaded, and mRNA expression values were analyzed with GraphPad (GraphPad Software, San Diego, CA, USA).

### 4.2. Endometrial Cancer Cell Lines

Ishikawa and MFE-280 cells, respectively well and poorly differentiated type I cell lines, were purchased from Sigma-Aldrich (Milan, Italy). Ishikawa cells were grown in EMEM medium (Lonza, Milan, Italy), supplemented with 5% fetal bovine serum (FBS), 2 mM/L glutamine, 100 IU/mL penicillin, 100 mg streptomycin. MFE-280 cells were grown in EMEM medium (Lonza, Milan, Italy), supplemented with 10% fetal bovine serum (FBS), 2 mM/L glutamine, 100 IU/mL penicillin, 100 mg streptomycin. HEC-1A and the primary endometrial cancer cell lines PCEM002, PCEM004a, PCEM004b were kindly provided by Dr. Fréderic Amant (Department of Oncology, KU Leuven, Leuven, Belgium). HEC-1A moderately differentiated type I cells were grown in McCoy’s Medium (Lonza, Milan, Italy), supplemented with 10% FBS, 100 IU/mL penicillin, 100 mg streptomycin, while all the primary cell lines were grown in RPMI1640, supplemented with 20% FBS, 2 mM/L glutamine, 100 IU/mL penicillin, 100 mg streptomycin. PCEM002 is a poorly differentiated type I cell line while PCEM004a and PCEM004b are poorly differentiated mixed type I/II cell lines. All cell lines were maintained at 37 °C with 5% CO_2_ and 95% humidity.

### 4.3. Materials

Pure CBD was supplied from ENECTA (Amsterdam, Netherlands). CBD was dissolved in ethanol. Chemotherapeutic drugs CIS, DOX and PAC were purchased from Sigma-Aldrich (Milan, Italy).

### 4.4. RNA Isolation, Reverse Transcription and Quantitative Real-Time PCR

Total RNA from cell lines was extracted with the RNeasy Mini Kit (Qiagen, Milan, Italy), and cDNA was synthesized using the iScript Advanced cDNA Synthesis Kit for RT-qPCR (Bio-Rad, Milan, Italy) according to manufacturer’s protocol. Quantitative real-time polymerase chain reactions (qRT-PCR) were performed with QuantiTect Primer Assays for human transient receptor potential vanilloid receptor 2 cation channel (*TRPV2*) and glyceraldehydes-3-phosphate dehydrogenase (*GAPDH*) (Qiagen, Milan, Italy), using the iQ5 Multicolor Real-Time PCR Detection System (Bio-Rad, Milan, Italy). The PCR parameters were 10 min at 95 °C followed by 40 cycles at 95 °C for 15 s and 60° C for 40 s. The relative amount of target mRNA was calculated by the 2^-ΔΔCt^ method, using *GAPDH* as a housekeeping gene. All samples were assayed in triplicates in the same plate. Measurement of *GAPDH* levels was used to normalize mRNA contents and target gene level was calculated by the 2^-ΔΔCt^ method.

### 4.5. Western Blot Analysis

20 ug of the lysates were separated on a sodium dodecyl sulfate (SDS) polyacrylamide gel, transferred onto Hybond-C extra membranes (GE Healthcare, Milan, Italy) and blocked with 5% low-fat dry milk in PBS-Tween 20 (Sigma-Aldrich). Immunoblots were incubated with goat polyclonal anti-TRPV2 (1:300, Santa Cruz Biotechnology, Dallas, TX, USA), rabbit polyclonal anti-LC3 (2 μg/mL, Novus Biologicals, Centennial, CO, USA), rabbit anti-pAKT (1:1000, Cell Signaling Technology, Danvers, MA, USA), anti-AKT (1:1000, Cell Signaling Technology) and rabbit anti-glyceraldehydes-3-phosphate dehydrogenase (GAPDH, 1:8000, OriGene Technologies, Rockville, MD, USA) antibodies (Abs) for 1 h and then with HRP-conjugated anti-mouse or anti-rabbit secondary Abs (1:2000, Cell Signaling Technology, Danvers, MA, USA) and with HRP-conjugated anti-goat secondary Ab (1:1000, Cell Signaling Technology, Danvers, MA, USA) for 1 h. Peroxidase activity was visualized with the LiteAblot^®^PLUS or TURBO (EuroClone, Milan, Italy) kit and densitometric analysis was carried out by a Chemidoc using the Quantity One software (Bio-Rad, Milan, Italy).

### 4.6. Patient Samples

After obtaining approval from the Medical Ethics Committee UZ/KU Leuven (protocol nr S61970, Dec 2018), 68 archived formalin-fixed, paraffin-embedded type II endometrial cancer samples, along with clinical data, and 15 normal tissues (5 of which, peritumoral tissues) were retrieved from UZ Leuven Biobank, Belgium. The sample set included 29 serous type tumors, 7 clear cell type tumors, 17 mixed type I and II, 5 peritumoral tissues from patients with type II endometrial cancer and 10 healthy endometrial tissue samples.

### 4.7. Immunohistochemical Stainings

Paraffin slides (4 μm) were heated for 3 to 4 h at 55 °C, deparaffinized in toluene, and rinsed in ethanol. Tissues were blocked for endogenous peroxidases by incubating them 30 min in 0.5% H_2_O_2_ (Merck Millipore, Burlington, MA, USA) in methanol. For TRPV2 staining, after washing in TBS, epitopes were retrieved for 1 h at 90 °C in Tris-sodium citrate (pH = 6). Tissues were cooled down slowly in TBS. After extensive washing, tissues were blocked with 1% milk powder, 2% BSA (Sigma-Aldrich), and 0.1% Tween-80 (Merck Millipore, Burlington, MA, USA) in TBS before antibody incubation. Blocking solutions were removed and tissues were incubated with rabbit anti-TRPV2 (VRL-1, 1:250, Thermofisher, Grand Island, NY, USA) in TBS, overnight at 4 °C. After washing, slides were incubated with anti-rabbit-HRP (Dako, Glostrup, Denmark) for 30 min and washed again. All antibodies were visualized by 10 min incubation in 3,3′-diaminobenzidine (DAB, Sigma-Aldrich, Milan, Italy) + 0.015% H_2_O_2_ in the dark. Nuclei were counterstained with Mayer’s hematoxylin, and tissues were dehydrated in propanol, dipped in xylene, and mounted. To ensure that no staining was caused by aspecific binding of secondary/tertiary molecules, control slides without addition of primary antibody were used.

### 4.8. Evaluation and Scoring of Immunohistochemical Stainings

All staining were evaluated semi-quantitatively, using an Allred scoring system that takes into account both intensity (0 = absent, 1 = weak, 2 = moderate, and 3 = strong) and percentage of stained cells (0 = absent, 1 = less than 1%, 2 = 1–10%, 3 = 11–33%, 4 = 34–66%, and 5 = 67–100%) [56]. Both scores were added to obtain a maximum score of 8. Tissues were considered with a high expression at a cut-off score of 6, corresponding to strong positivity in ≥11% of cells, moderate positivity in ≥34% of cells, or weak staining in ≥67% of cells. This cut-off was deemed clinically relevant for therapeutic applications, as a targeted therapy would most likely be effective when a sufficient number of cells express the target. Tissues with a value between 4 and 5 were classified as moderate. Photographs of representative cases were taken using the Axioskop microscope (MRc5, Zeiss, Oberkochen, Germany) and the ZEN 2.0 software.

### 4.9. Cell Transfection

Ishikawa cells were plated at density 3 × 10^4^ cells/mL. After 12 h incubation, transfection was performed with 3 μl/mL of the reagent TurboFectin Transfection Reagent (OriGene Technologies, Rockville, MD, USA) and 1 μg/mL of pCMV empty (pCMV6) or pCMV6-TRPV2 vector (OriGene Technologies), according to manufacturer’s instructions. Cells were harvested at 72 h post-transfection for subsequent analyses. Transfection efficiency was evaluated by Western blot analysis.

### 4.10. Wound-Healing Assay

Cells were seeded onto a twelve-well plate at density 3 × 10^4^ cells/mL. Confluent cells were scratched using 10 µl sterile pipette tips. Complete medium was replaced with fresh medium supplemented with low percentage of serum to minimize cell proliferation and prevent cell detachment. Images of wounded areas were taken at 0 and 24 h for wild-type cells, treated with CBD at 3.92 and 7.85 µg/mL. Additionally, migration was evaluated in non-transfected and transfected cells for up 48 h. Images acquisition was carried out by a LeitzFluovert FU (Leica Microsystems, Wetzlar, Germany) microscope. Remaining wound areas were determined using NIH Image J software (Research Services Branch (RSB), National Institutes of Health (NIH), Bethesda, MD, USA) for calculation of the percentage of wound closure. Analyses were performed in triplicate.

### 4.11. 3-[4,5-Dimethylthiazol-2-Yl]-2,5 Diphenyl Tetrazolium Bromide (MTT) Assay

Cell lines (3 × 10^4^ cells/mL) were seeded in 96-well plates, in a final volume of 100 μL/well. After one day of incubation, CBD was added in single or daily administration for 72 h, after washing with fresh medium, while chemo-drugs were single administered. At least six replicates were used for each treatment. At the indicated time point, cell viability was assessed by adding 0.8 mg/mL of 3-[4,5-dimethylthiazol-2-yl]-2,5 diphenyl tetrazolium bromide (MTT) (Sigma-Aldrich, Milan, Italy) to the media. The absorbance of samples, solubilized in dimethyl sulfoxide (DMSO), against a background control (medium alone) was measured at 570 nm using a reader microliter plate (BioTek Instruments, Winooski, VT, USA).

### 4.12. Apoptosis Assays and PI Staining

Cell death was evaluated using Annexin V-FITC Apoptosis detection Kit (eBioscience, Milan, Italy) followed by biparametric FACS analysis. Cells, at a density of 3 × 10^4^ cells/mL, were treated with CBD for a maximum of 72 h, in daily administration, and then incubated with Annexin V-FITC and PI, following manufacturer’s protocol. Percentage of positive cells determined over 10,000 events was analyzed on a FACScan cytofluorimeter using the CellQuest software (BD Biosciences, San Jose, CA, USA).

### 4.13. Cell Cycle Analysis

Cells, at a density of 3 × 10^4^ cells/mL, were treated with CBD for up to 48 h, in daily administration. Cells were fixed by adding ice-cold 70% ethanol for 1 h and then washed with staining buffer (PBS, 2% FBS and 0.01% NaN_3_). Next, 100 µg/mL ribonuclease A solution (Sigma-Aldrich, Milan, Italy) was added for 30 min at 37 °C, stained with 20 µg/mL propidium iodide (PI) (Sigma-Aldrich, Milan, Italy) for 30 min at room temperature and finally analyzed by flow cytometry using linear amplification.

### 4.14. Acridine Orange Staining

To detect the development of acidic vesicular organelles, which are the hallmarks of autophagy, the vital staining of PCEM004 cells with acridine orange (AO, Sigma-Aldrich, Milan, Italy) was performed. The cells, at density 3 × 10^4^ cells/mL, were seeded in 12-well plates and were treated with CBD at 3.92 and 7.85 µg/mL for 48 h. Then, cells were stained with medium containing 1 μg/mL AO for 15 min at 37 °C, washed twice in PBS and immediately examined with Nikon Eclipse E800 fluorescence microscope and NIS-Elements 4.0 software (Nikon, Tokyo, Japan). Cytoplasm and nuclei of AO-stained cells fluoresced bright green, whereas the acidic autophagic vacuoles fluoresced bright red.

### 4.15. Statistical Analysis

The data presented represent the mean with standard deviation (SD) of at least 3 independent experiments. The statistical significance was determined by Student’s t-test and by One Way-Anova and Two Way-Anova with Bonferroni’s post-test; * *p* < 0.05. Patients were divided in three groups according to high, moderate or low expression of the target protein. The Kaplan–Meier (KM) method was used for Overall Survival and Progression-Free Survival analysis. For Univariate analysis of significance (GraphPad Software, San Diego, CA, USA), long-rank test was used. * *p* < 0.05 was considered as statistically significant. Statistical analysis of IC_50_ levels was performed using Prism 5.0a (GraphPad Software, San Diego, CA, USA).

## 5. Conclusions

In conclusion, TRPV2 could be considered as a new potential marker for type II EC, especially serous subtype and high-grade tumors. In addition, its expression increased EC aggressiveness, enhancing migratory capacity through AKT/mTOR activation. CBD, a TRPV2 ligand, might be a potentially useful therapeutic option as adjuvant therapy to increase the efficacy of chemotherapeutic drugs to reduce cancer cell spreading.

## Figures and Tables

**Figure 1 ijms-21-05409-f001:**
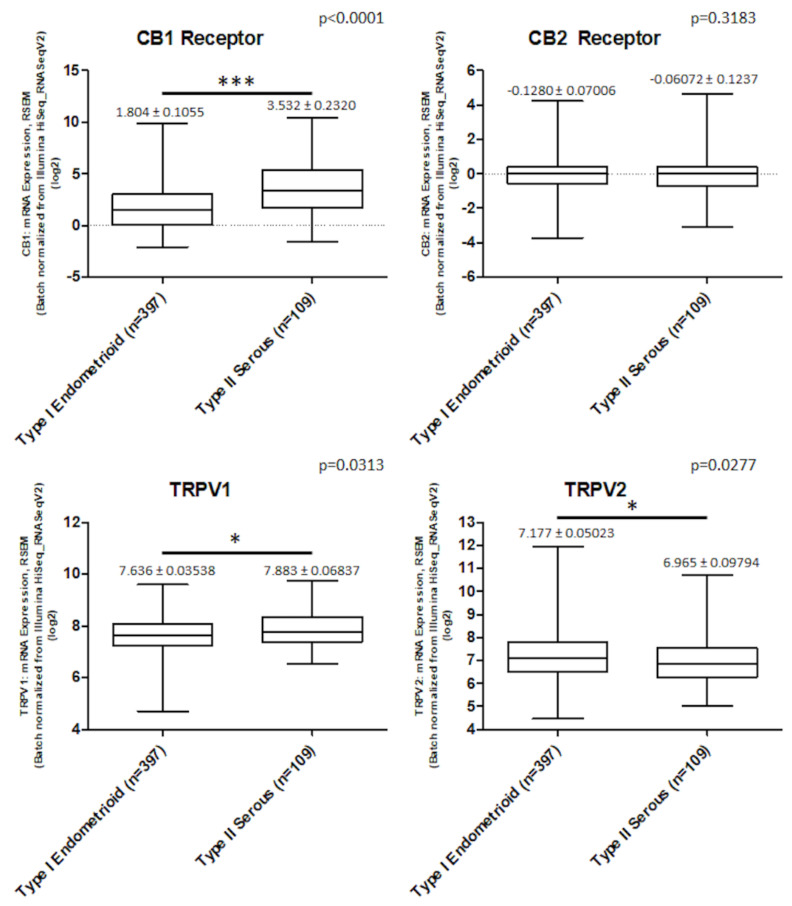
The expression of CBD (cannabidiol) targets in EC (endometrial cancer) patients. The mRNA expression (log RNA Seq V2 RSEM) of *CB1*, *CB2*, *TRPV1* and *TRPV2* in 506 EC samples, divided in 397 for type I and 109 for type II, from TCGA database. *** *p* < 0.001 type II vs. type I, * *p* < 0.05 type II vs. type I.

**Figure 2 ijms-21-05409-f002:**
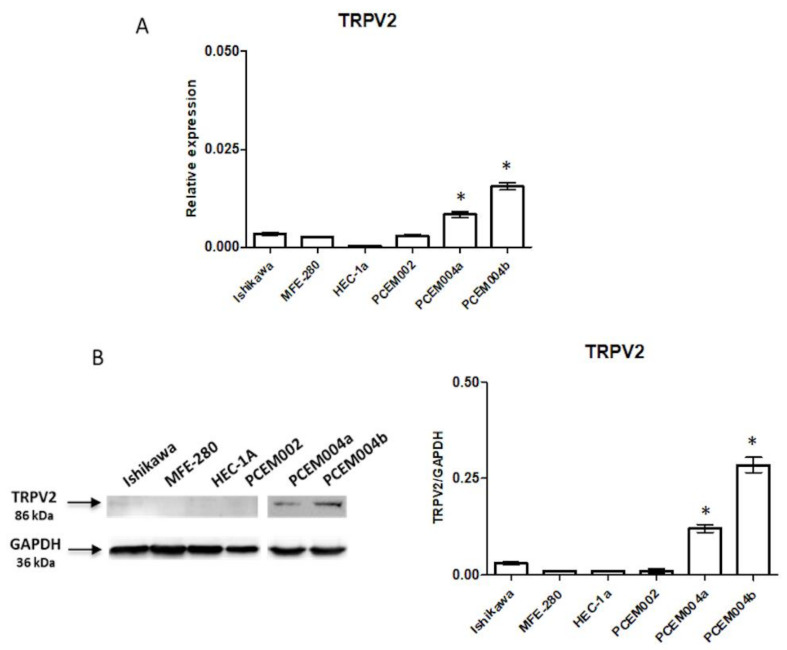
TRPV2 expression on EC cell lines. (**A**) *TRPV2* mRNA expression was evaluated by quantitative real time-PCR (qRT-PCR) in six EC cell lines. *TRPV2* mRNA levels were normalized for glyceraldehyde-3-phosphate dehydrogenase (*GAPDH)* expression. Data are expressed as fold mean ± standard deviation (SD) of three separate experiments. * *p* < 0.05 vs. type I EC cell lines (**B**) TRPV2 protein expression was evaluated by Western blot in six EC cell lines. TRPV2 densitometry values were normalized to GAPDH used as loading control. Densitometric values shown are the mean ± SD of three separate experiments. * *p* < 0.05 vs. type I EC cell lines.

**Figure 3 ijms-21-05409-f003:**
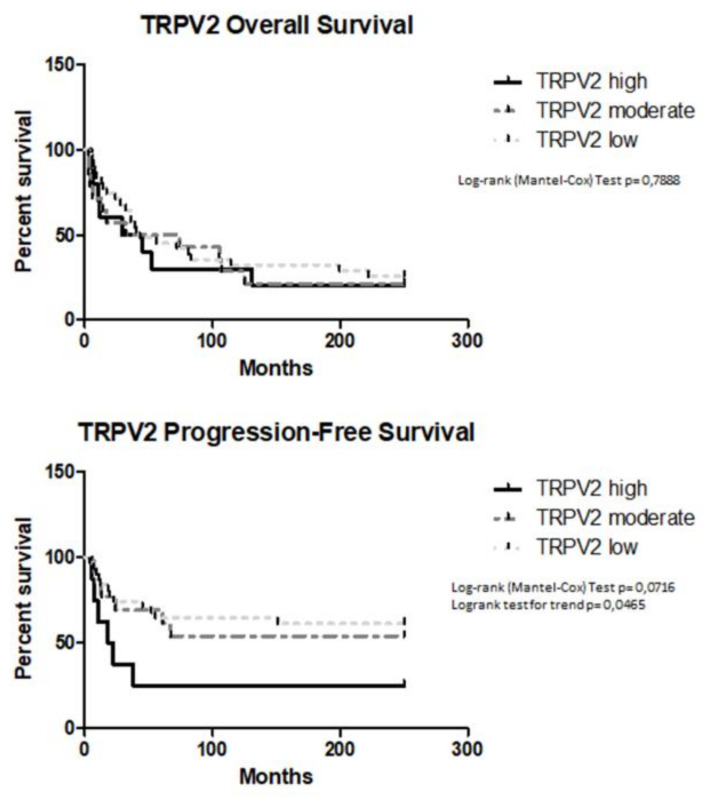
Survival of EC patients according to TRPV2 expression. Kaplan–Meier survival curves showing OS (overall survival) and PFS (progression-free survival) of EC patients. The log-rank test with corresponding P values applies to the TRPV2—high, TRPV2—moderate and TRPV2—low curves.

**Figure 4 ijms-21-05409-f004:**
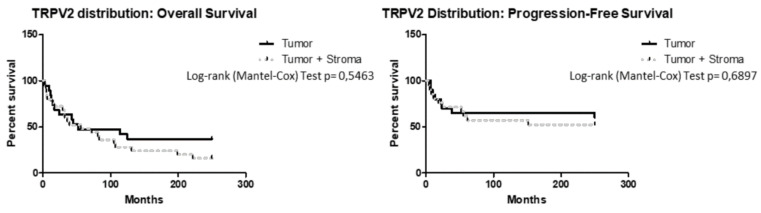
Survival of EC patients according to TRPV2 distribution. Kaplan–Meier survival curves showing OS and PFS of EC patients, according to TRPV2 distribution inside tumoral mass. The log-rank test with corresponding *p* values applies to the TRPV2—tumor and TRPV2—tumor + stroma.

**Figure 5 ijms-21-05409-f005:**
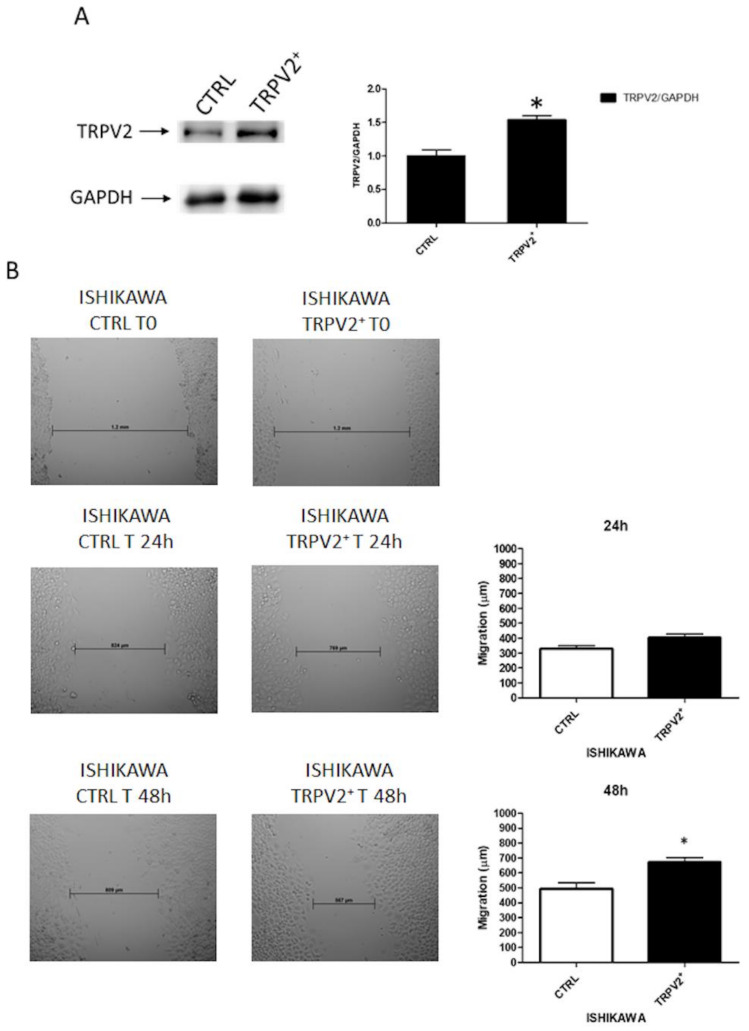
TRPV2 over-expression enhanced migration of EC cells. (**A**) Western blot analysis of TRPV2 and GAPDH protein levels in untransfected (CTRL) and TRPV2-transfected Ishikawa cells. Blots are representative of one of three separate experiments. TRPV2 densitometry values were normalized to GAPDH used as loading control and normalized. Densitometric values shown are the mean ± SD of three separate experiments. * *p* < 0.05 vs. control cells. (**B**) Wound-healing assays for Ishikawa cells after TRPV2 over-expression. All experiments were repeated three times. Data are presented as the mean ± SD. * *p* < 0.05 vs. control.

**Figure 6 ijms-21-05409-f006:**
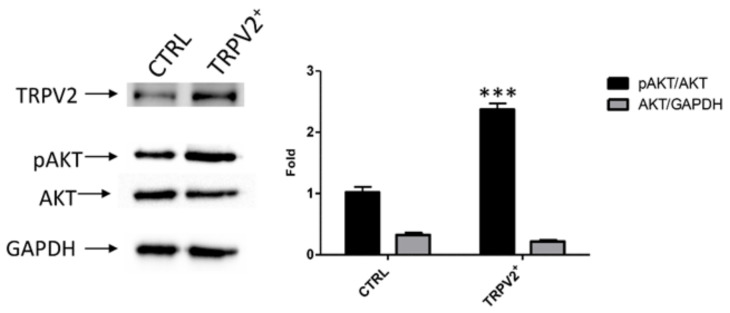
TRPV2 over-expression increased phosporilated form of AKT (pAKT) in EC cells. Representative Western blot analysis of pAKT(Ser473), AKT and GAPDH protein levels in TRPV2^+^ cells. The pAKT(Ser473) protein levels were determined with respect to AKT levels. AKT densitometry values were normalized to GAPDH used as loading control. Densitometric values shown are the mean ± SD of three separate experiments. *** *p* < 0.001 vs. control cells.

**Figure 7 ijms-21-05409-f007:**
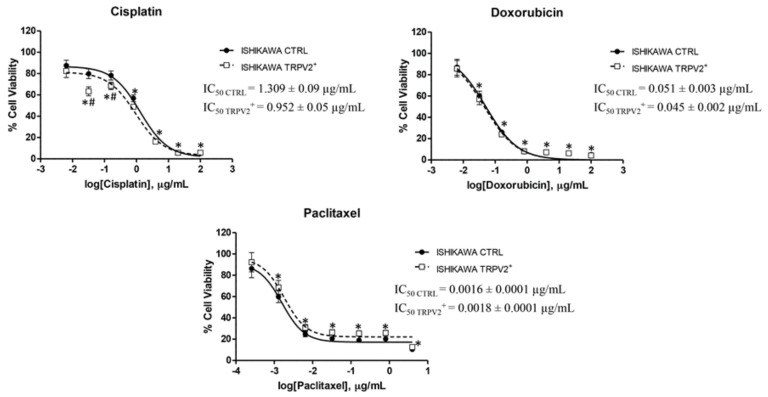
TRPV2 over-expression enhanced CIS sensitivity. Untransfected and TRPV2^+^ cells were treated for 72 h with different concentrations of CIS, DOX (up to 100 μg/mL) and PAC (up to 4 μg/mL). Data shown are expressed as mean ± SD of three separate experiments. * *p* < 0.05 treated vs. vehicle, # *p* < 0.05 TRPV2^+^ vs. control.

**Figure 8 ijms-21-05409-f008:**
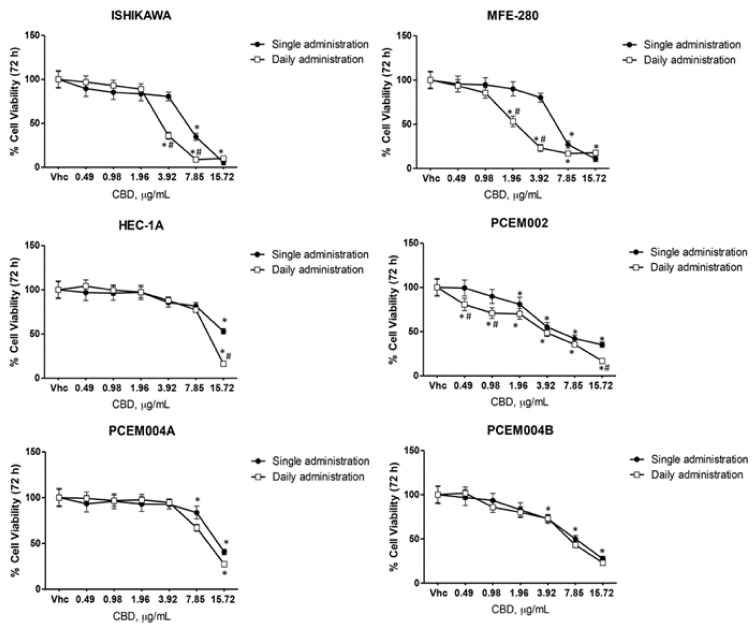
CBD induced cytotoxicity in EC cell lines. Cell viability was determined by MTT assay. Ishikawa, MFE-280, HEC-1a, PCEM002, PCEM004a and PCEM004b cells were treated for 72 h with different concentrations of CBD (up to 15.72 µg/mL). Data shown are expressed as mean ± SD of three separate experiments. * *p* < 0.05 treated vs. vehicle, # *p* < 0.05 daily vs. single administration.

**Figure 9 ijms-21-05409-f009:**
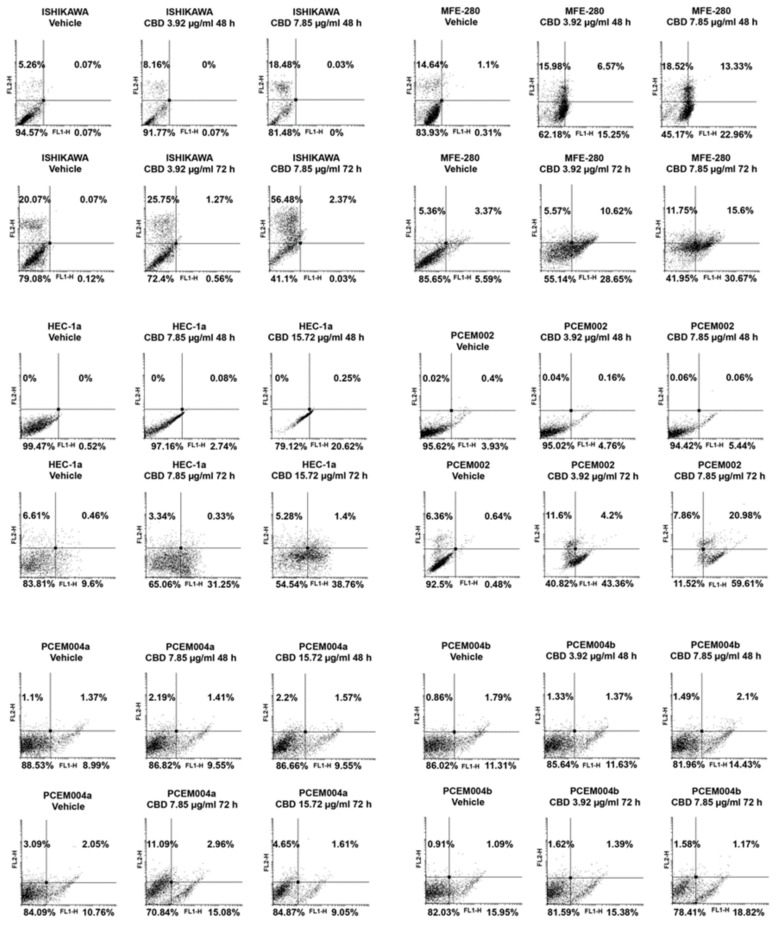
CBD induced cell death in EC cell lines. EC cell lines were treated with CBD for 72 h. Flow cytometric analysis was performed by Annexin V and PI double-staining. FL1-H fluorescence signal is in *x*-axis. Data represent the percentage of PI and/or Annexin V^+^ cells and are representative of one of three separate experiments.

**Figure 10 ijms-21-05409-f010:**
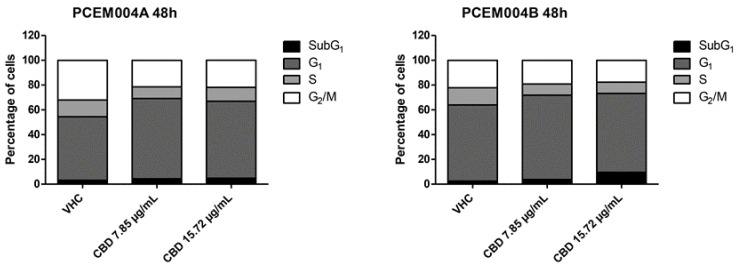
CBD induced cell cycle arrest in PCEM004 EC cell lines. PCEM004a and PCEM004b EC cell lines were treated with CBD up to 15.72 µg/mL for 48 h. Percentage of cells are represented as histogram representations of the cell cycle phases in the EC cell lines and are representative of one of three separate experiments.

**Figure 11 ijms-21-05409-f011:**
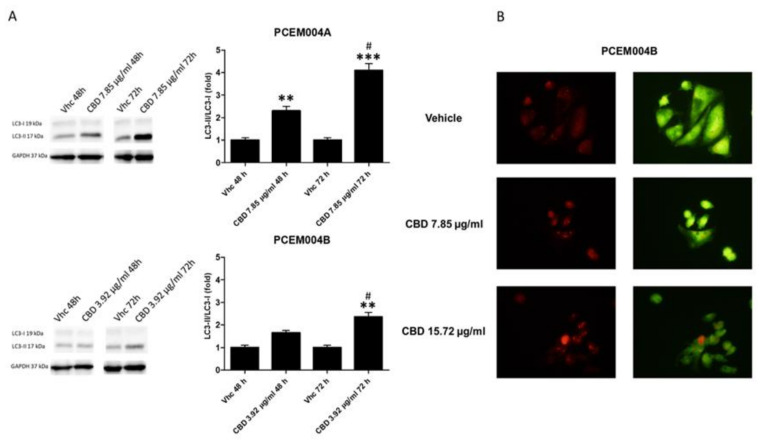
CBD induced autophagy in mixed type I/II EC cell lines. (**A**) Western blot analysis and densitometric quantification of LC3 protein levels PCEM004a and PCEM004b mixed type EC cell lines treated for up to 72 h with CBD 7.85 µg/mL and 3.92 µg/mL, respectively. Densitometric values, shown as the mean ± SD, were normalized to GAPDH used as loading control. Blots are representative of one of three separate experiments, ** *p* < 0.01, *** *p* < 0.001 treated vs. untreated cells, # *p* < 0.05 72 h vs. 48 h of treatment. (**B**) Representative image of PCEM004a treated with CBD up to 7.85 µg/mL for 48 h and stained with acridine orange.

**Figure 12 ijms-21-05409-f012:**
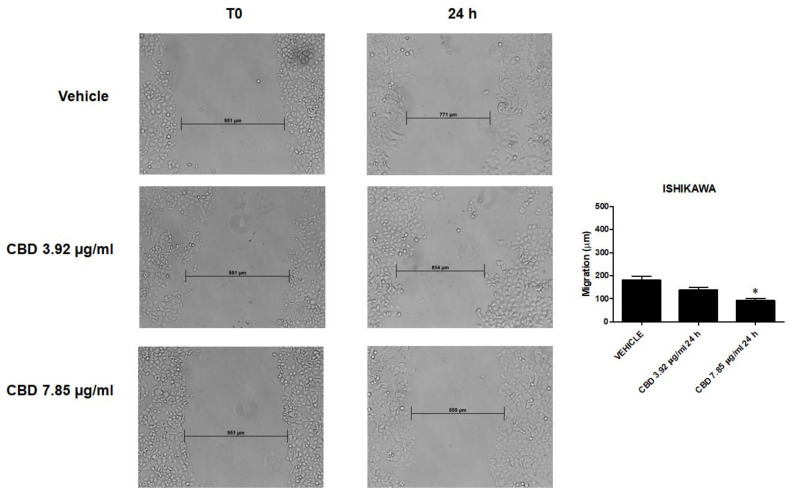
CBD treatment inhibited migration of Ishikawa cells. Wound-healing assays for Ishikawa cells after treatment with CBD 7.85 µg/mL and 3.92 µg/mL for 24 h. All experiments were repeated three times and images were taken at 0 and 24 h (20×). Data are presented as the mean ± SD. * *p* < 0.05 vs. untreated.

**Figure 13 ijms-21-05409-f013:**
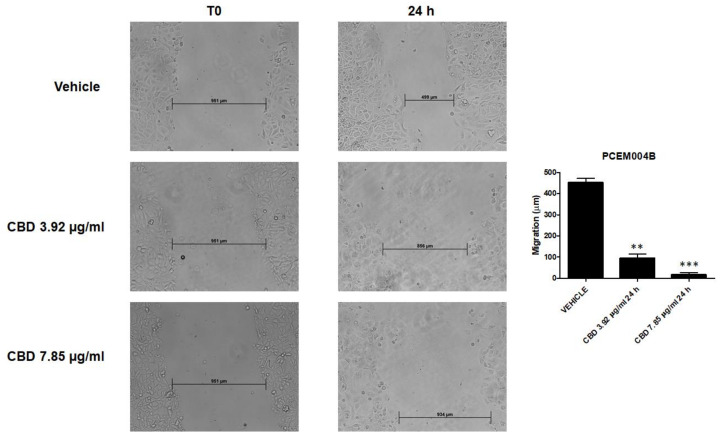
CBD treatment inhibited migration of PCEM004b cells. Wound-healing assays for PCEM004b cells after treatment with CBD 7.85 µg/mL and 3.92 µg/mL for 24 h. All experiments were repeated three times and images were taken at 0 and 24 h (20×). Data are presented as the mean ± SD. ** *p* < 0.01, *** *p* < 0.001 vs. untreated.

**Figure 14 ijms-21-05409-f014:**
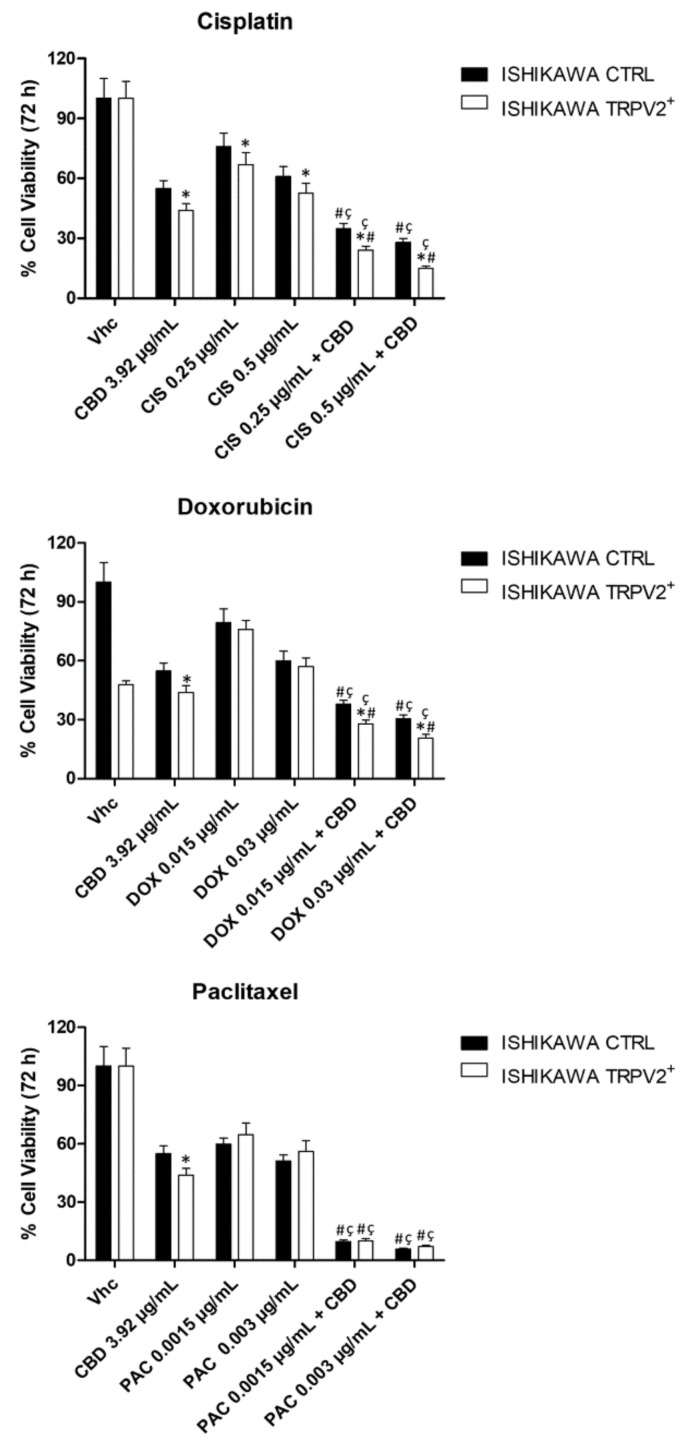
CBD improved chemotherapeutic drug effects in TRPV2-transfected Ishikawa cell line. Cell viability was determined in EC cell lines by MTT assay. Cells were treated for 72 h with CBD, alone and in combination with different doses of CIS, DOX and PAC. Data shown are expressed as mean ± SD of three separate experiments. * *p* < 0.05 TRPV2^+^ vs. control, # *p* < 0.05 vs. CBD alone, ç *p* < 0.05 vs. chemotherapeutic drug alone.

**Table 1 ijms-21-05409-t001:** Expression of TRPV2 in EC biopsies according to different clinicopathological characteristics, in EC biopsies, peritumoral tissue and normal endometrium. Percentages of samples positive for TRPV2 expression are shown.

	TRPV2		
	High	Moderate	Low
**Tumor**	10/53 (18.86%)	16/53 (30.19%)	27/53 (50.94%)
Serous	7/29 (24.13%)	9/29 (31.03%)	13/29 (45.89%)
Clear cell	1/7 (14.28%)	1/7 (14.28%)	5/7 (71.43%)
Mixed	2/17 (11.76%)	6/17 (35.29%)	9/17 (52.94%)
**Peritumoral tissue**	0/5 (0%)	1/5 (20%)	4/5 (80%)
**Normal endometrium**	0/10 (0%)	4/10 (40%)	6/10 (60%)
**FIGO stage**			
**Stage I–II**	3/20 (15%)	5/20 (25%)	12/20 (60%)
Serous	1/10 (10%)	3/10 (30%)	6/10 (60%)
Clear cell	0/2 (0%)	1/2 (50%)	1/2 (50%)
Mixed	2/8 (25%)	1/8 (12.5%)	5/8 (62.5%)
**Stage III**	3/17 (17.65%)	6/17 (35.29%)	8/17 (47.06%)
Serous	3/10 (30%)	4/10 (40%)	3/10 (30%)
Clear cell	0/2 (0%)	0/2 (0%)	2/2 (100%)
Mixed	0/5 (0%)	2/5 (40%)	3/5 (60%)
**Stage IV**	4/16 (25%)	5/16 (31.25%)	7/16 (43.75%)
Serous	4/9 (44.44%)	2/9 (22.22%)	3/9 (33.33%)
Clear cell	1/3 (33.33%)	0/3 (0%)	2/3 (66.66%)
Mixed	0/4	3/4 (75%)	1/4 (25%)
**Age**			
≤68	4/26 (15.38%)	6/26 (23.07%)	16/26 (61.54%)
>68	6/27 (22.22%)	10/27 (37.04%)	11/27 (40.74%)

**Table 2 ijms-21-05409-t002:** CBD IC_50_ values in EC cell lines expressed as mean ± SD of three separate experiments.

	IC_50_ CBD µg/mL		
	Single Administration	Daily Administration	*p* Value
Ishikawa	5.89 ± 0.4	3.56 ± 0.2	**
MFE-280	6.33 ± 0.5	2.39 ± 0.1	**
HEC-1A	23.38 ± 0.8	13.16 ± 0.6	*
PCEM002	6.58 ± 0.2	3.59 ± 0.1	**
PCEM004a	20.02 ± 0.7	14.01 ± 0.6	*
PCEM004b	8.29 ± 0.5	7.05 ± 0.4	

* *p* < 0.05, ** *p* < 0.01, daily vs. single administration.

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
