# Peer review of "The Effects of Cannabidiol and Prognostic Role of TRPV2 in Human Endometrial Cancer"

_ijms, 2020, doi:10.3390/ijms21155409_

Round 1

Reviewer 1 Report

The authors here evaluated the expression of TRPV2 in Endometrial cancer biopsies and cell lines and possible use of CBD, a TRPV2 ligand, as adjuvant therapy. The manuscript is well written with adequate references and quite a comprehensive set of experimental evidence. However, I do have some concerns and comments –

  • I would suggest to rephrase this sentence – Line 37 – “The CBD improved chemotherapeutic drugs cytotoxic effects, enhanced by TRPV2 over-expression.”
  • The resolution of Fig 2B is poor and TRPV2 levels are almost negligible in 4 of the 6 cell lines tested. However, the Ishikawa control in both Figure 5A and 6 show a much stronger band. Unclear as to why there is this discrepancy.
  • In line 169 – “we decided to investigate the role of TRPV2 on proliferation and migration of Ishikawa cells, no expressing TRPV2 expression.” This sentence needs to be rephrased. Currently, it seems to be implying that Ishikawa has no TRPV2 expression, when in fact Figure 5A and Figure 6 clearly shows low but non-zero expression of TRPV2 in control samples.
  • Fig 1 showed that TRPV2 expression is almost negligible in Ishikawa, MFE-280, HEC-1A and PCEM002 cell lines. However, treatment with CBD a TRPV ligand showed significant cytotoxicity in all four cell lines in the MTT assay (Figure 8). In fact it showed higher potency in Ishikawa, MFE-280 and PCEM002 cells than PCEM004a cells that showed much higher expression of TRPV2 (Figure 1B). Wouldn’t this suggest a non-TRPV2 mechanism of action? This also brings into question the major conclusion of the manuscript about the possible use of CBD in endometrial cancer via TRPV2 activity. Some additional discussion about this would be helpful.
  • Line 223 – “For this reason, we decided to deepen CBD effects in PCEM004a and PCEM004b mixed type EC cells.” should probably be rephrased.
  • There is quite a big difference in migration in the wound healing assay between Control in Figure 5 (~350 µm) and Vehicle in Figure 12 (~200 µm) for the same cell line. Is this due to the addition of vehicle or just experimental noise?
  • In line 258, it is stated that both PCEM004a and b cell lines were tested for wound healing assay with CBD. However, only the PCEM004b data is provided in Figure 13. In general, the wound-healing assay pictures could be improved. The resolution of the images in the reviewer version are quite poor, especially Figure 5.

Overall, the paper, although comprehensive, reads like an aggregation of a number of experiments with different cell lines and drug compounds. It is a bit difficult to follow the rationale behind why certain cell lines were chosen for certain experiments and immediately switched to a different cell line for the next set of experiments.

Author Response

RESPONSE TO REVIEWERS:

Reviewer 1

Comments and Suggestions for Authors

The authors here evaluated the expression of TRPV2 in Endometrial cancer biopsies and cell lines and possible use of CBD, a TRPV2 ligand, as adjuvant therapy. The manuscript is well written with adequate references and quite a comprehensive set of experimental evidence. However, I do have some concerns and comments –

Q: I would suggest to rephrase this sentence – Line 37 – “The CBD improved chemotherapeutic drugs cytotoxic effects, enhanced by TRPV2 over-expression.”

A: Line 37: we modified the sentence

Q: The resolution of Fig 2B is poor and TRPV2 levels are almost negligible in 4 of the 6 cell lines tested. However, the Ishikawa control in both Figure 5A and 6 show a much stronger band. Unclear as to why there is this discrepancy.

A: In blot presented in fig. 2 (in which Ishikawa cells showed a very low expression of TRPV2) we used a strong positive control for TRPV2. In order to avoid a saturation for the positive control, the intensity of Ishikawa band is lower than blots showed in figure 5 and 6, in which we omitted the positive control, as shown in whole blots images in attachments.

Q: In line 169 – “we decided to investigate the role of TRPV2 on proliferation and migration of Ishikawa cells, no expressing TRPV2 expression.” This sentence needs to be rephrased. Currently, it seems to be implying that Ishikawa has no TRPV2 expression, when in fact Figure 5A and Figure 6 clearly shows low but non-zero expression of TRPV2 in control samples.

A: Lines 168: the sentence has been correct. Ishikawa cell line shows a low trpv2 expression.

Q: Fig 1 showed that TRPV2 expression is almost negligible in Ishikawa, MFE-280, HEC-1A and PCEM002 cell lines. However, treatment with CBD a TRPV ligand showed significant cytotoxicity in all four cell lines in the MTT assay (Figure 8). In fact, it showed higher potency in Ishikawa, MFE-280 and PCEM002 cells than PCEM004a cells that showed much higher expression of TRPV2 (Figure 1B). Wouldn’t this suggest a non-TRPV2 mechanism of action? This also brings into question the major conclusion of the manuscript about the possible use of CBD in endometrial cancer via TRPV2 activity. Some additional discussion about this would be helpful.

A: Lines 367-369: we added in the discussion the statement “However, since it is known that CBD acts both in a dependent (D)- and independent (ID)-receptors mechanisms (RMs) [26-28], our results support that CBD effects can be obtained also in TRPV2-low expressing EC cells. “ 

Q: Line 223 – “For this reason, we decided to deepen CBD effects in PCEM004a and PCEM004b mixed type EC cells.” should probably be rephrased.

A: Line 224: we added “investigate” in the sentence

Q: There is quite a big difference in migration in the wound healing assay between Control in Figure 5 (~350 µm) and Vehicle in Figure 12 (~200 µm) for the same cell line. Is this due to the addition of vehicle or just experimental noise?

A: In agreement with the reviewer, the difference could be due to the addition of vehicle (EtOH), while in figure 5 it has been analyzed the effect of TRPV2 over-expression compared with control in only medium without EtOH.  

Q: In line 258, it is stated that both PCEM004a and b cell lines were tested for wound healing assay with CBD. However, only the PCEM004b data is provided in Figure 13. In general, the wound-healing assay pictures could be improved. The resolution of the images in the reviewer version are quite poor, especially Figure 5.

A: Results of wound-healing assay for PCEM004a are added as Supplementary Figure 2 and the overall quality of images has been improved.

Q: Overall, the paper, although comprehensive, reads like an aggregation of a numbers of experiments with different cell lines and drug compounds. It is a bit difficult to follow the rationale behind why certain cell lines were chosen for certain experiments and immediately switched to a different cell line for the next set of experiments.

A: We evaluated the expression of TRPV2 and CBD effect in all our EC cell lines, both Type I and mixed type I/II. Then, we selected a model of type I (Ishikawa) with low expression of TRPV2 and all the mixed type cells (PCEM004a and b) with higher TRPV2 expression compared to Ishikawa cells. Our paper focused on the TRPV2 role in both types of EC. CBD was used since it is one of most established TRPV2 activator and it also considered a cannabinoid compound with a TRPV2-dependent and independent antitumoral activity. Summarising, our data evidenced the TRPV2 expression is associated to aggressiveness; TRPV2 activation by CBD increases chemosensitivity and CBD can exert an antitumoral effect also in a TRPV2-independent manner as reported in different other papers.

Reviewer 2 Report

The manuscript titled “The effects of cannabidiol and prognostic role of TRPV2 in human endometrial cancer” deals with the up-to-date topic – endometrial cancer.

The manuscript is written and designed well. However, the manuscript has few shortcomings. The abbreviations are due to “non-standard” organization of the manuscript (Materials and methods at the end of the manuscript) not explained at the first time of their use. Please, correct it. The manuscript contains few grammatical and stylistical errors, please discuss the manuscript with the native speaker (f. e. line 60 …D-RMs mainly includes…). The names of the proteins are written in standard font, however, the names of the genes and their mRNA should be written in italics. Please, correct it.

Figure 1 includes some significant parameters, however, from the legend of the figure, it is not clear, if the data include SEM or SD (please, add the information). However, the statistics seems to be weird, because the SD/SEM seem to be overlapping, thus, there could be no statistical difference. Please, give the data into the table.

How could you explain that you have found no CB2 receptors in EC patients? There are some information that EC patients do express CB2 (https://pubmed.ncbi.nlm.nih.gov/30569804/, https://www.frontiersin.org/articles/10.3389/fonc.2019.00430/full).

Other figures contain SE. Should it be SEM or SD? Please, correct it.

Figure 8 contains the information about single or daily administration of CBD. However, I miss the information about the method of the dosing. The daily administration means that you added to the medium 3x CBD? Or have you changed the medium every day and gave there a new dose of CBD? It is not clear from the text. Please specify.

In the discussion part, line 331, there are two forms of the citations (Nabissi et al., 2013). Please, correct it.

This study is interesting and has an impact on scientific community. After minor revisions I would recommend this manuscript to be accepted to the International Journal of Molecular Sciences.

Author Response

Reviewer 2

Comments and Suggestions for Authors

The manuscript titled “The effects of cannabidiol and prognostic role of TRPV2 in human endometrial cancer” deals with the up-to-date topic – endometrial cancer.

Q: The manuscript is written and designed well. However, the manuscript has few shortcomings. The abbreviations are due to “non-standard” organization of the manuscript (Materials and methods at the end of the manuscript) not explained at the first time of their use. Please, correct it. The manuscript contains few grammatical and stylistical errors, please discuss the manuscript with the native speaker (f. e. line 60 …D-RMs mainly includes…). The names of the proteins are written in standard font, however, the names of the genes and their mRNA should be written in italics. Please, correct it.

  1. Line 58: We corrected the sentence

Line 79, 216-217, 400, 472: we added and explained some abbreviations

We modified as requested, the gene names with the standard font.

Q: Figure 1 includes some significant parameters, however, from the legend of the figure, it is not clear, if the data include SEM or SD (please, add the information). However, the statistics seem to be weird, because the SD/SEM seems to be overlapping, thus, there could be no statistical difference. Please, give the data into the table.

R: Figure 1 was modified (histograms were replaced with box-plot graphs) as represented in cBioPortal with the means and range of expression (maximum and minimum expression values) and statistical analysis. All the single data are available at: http://cbioportal.org and the datasets can be downloaded as described in Mats& Meths.

Q: How could you explain that you have found no CB2 receptors in EC patients? There are some information that EC patients do express CB2 (https://pubmed.ncbi.nlm.nih.gov/30569804/, https://www.frontiersin.org/articles/10.3389/fonc.2019.00430/full).

R: According to CBioPortal data represented in new figure 1, patients showed a very low mean level of CB2 expression.

Q: Other figures contain SE. Should it be SEM or SD? Please, correct it.

R: We apologize for the mistake. All the data are represented as means ± SD

Q: Figure 8 contains the information about single or daily administration of CBD. However, I miss the information about the method of the dosing. The daily administration means that you added to the medium 3x CBD? Or have you changed the medium every day and gave there a new dose of CBD? It is not clear from the text. Please specify.

A: Line 201: we explain better in the text. In daily administration, we washed cells with fresh medium, and then we added one volume of medium plus one volume of treatment 2x every day.

  1. In the discussion part, line 331, there are two forms of the citations (Nabissi et al., 2013). Please, correct it.

A: Line 330: corrected

This study is interesting and has an impact on scientific community. After minor revisions I would recommend this manuscript to be accepted to the International Journal of Molecular Sciences.

Reviewer 3 Report

This paper is addressing an evolving topic, TRPV2 in cancer. The focus is on endometrial cancer and the potential role of cannadidiol. The authors performed a  study, which involved human material, patient data  and different cell lines. Overall the paper reads well.

However I have the following concerns:

1. Antibody specificity of TRPV2,  in particular: The Westerns shown in several figures reveal only a small area of the blot and only the apparently expected band. However, the whole blot must shown. Have the authors attempted to perform pre-adsorption controls? As mRNA and protein expression may differ this is an important point.

2. The authors do not mentioned (at least I can not find this information) how often the many experiments were repeated. They mention (in the legends) a that the data are mean +/- SE but without giving a clear “n“ the robustness and validity  can not be evaluated.

3. Immunohistochmical staining (p 18): Apparently there are controls missing (rabbit IgG, non-immune serum, pre-adsorped antibody). Omission of the antibody alone is not sufficient.

Round 2

Reviewer 3 Report

The points of concern were addressed.